# Broad-Spectrum Antibiotic Use and Disease Progression in Early-Stage Melanoma Patients: A Retrospective Cohort Study

**DOI:** 10.3390/cancers13174367

**Published:** 2021-08-29

**Authors:** Mahip Acharya, Thomas Kim, Chenghui Li

**Affiliations:** 1Division of Pharmaceutical Evaluation and Policy, University of Arkansas for Medical Sciences College of Pharmacy, Little Rock, AR 72205, USA; macharya@uams.edu; 2Department of Radiation Oncology, Rush University Medical College, Chicago, IL 60612, USA; Thomas_Kim@rush.edu

**Keywords:** melanoma, broad-spectrum antibiotics, progression, claims data

## Abstract

**Simple Summary:**

We conducted a retrospective cohort study to evaluate the association between broad-spectrum antibiotic use and disease progression in early-stage melanoma patients who underwent surgery. We used healthcare claims data (2008–2018) and identified individuals with melanoma diagnosis and surgery within 90 days of diagnosis. We studied the relationship between melanoma progression (a proxy measure created using cancer therapies, surgery, and metastasis) within 2 years of melanoma surgery and antibiotic use in three time windows separately: (i) 3 months prior to surgery, (ii) 1 month after surgery, and (iii) 3 months after surgery. We found that prescriptions for antibiotics in 3 months prior to surgery were not associated with melanoma progression; in contrast, antibiotic use in post-1- and post-3-months windows was associated with reduced risk of progression. Our study is exploratory and limited to early-stage melanoma patients with surgery.

**Abstract:**

Animal studies and a few clinical studies have reported mixed findings on the association between antibiotics and cancer incidence. Antibiotics may inhibit tumor cell growth, but could also alter the gut-microbiome-modulated immune system and increase the risk of cancer. Studies that assess how antibiotics affect the progression of cancer are limited. We evaluated the association between broad-spectrum antibiotic use and melanoma progression. We conducted a retrospective cohort study using IQVIA PharMetrics^®^ Plus data (2008–2018). We identified patients with malignant melanoma who underwent wide local excision or Mohs micrographic surgery within 90 days of first diagnosis. Surgery date was the index date. Patients were excluded if they had any other cancer diagnosis or autoimmune disorders in 1 year before the index date (“baseline”). Exposure to broad-spectrum antibiotics was identified in three time windows using three cohorts: 3 months prior to the index date, 1 month after the index date, and 3 months after the index date. The covariates were patients’ demographic and clinical characteristics identified in the 1-year baseline period. The patients were followed from the index date until cancer progression, loss of enrollment, or the end of 2 years after the index date. Progression was defined as: (i) any hospice care after surgery, (ii) a new round of treatment for melanoma (surgery, chemotherapy, immunotherapy, targeted therapy, or radiotherapy) 180 days after prior treatment, or (iii) a metastasis diagnosis or a diagnosis of a new nonmelanoma primary cancer at least 180 days after first melanoma diagnosis or prior treatment. A high-dimensional propensity score approach with inverse weighting was used to adjust for the patients’ baseline differences. Cox proportional hazard regression was used for estimating the association. The final samples included 3930, 3831, and 3587 patients (mean age: 56 years). Exposure to antibiotics was 16% in the prior-3-months, 22% in the post-1-month, and 22% in the post-3-months. In the pre-3-months analysis, 9% of the exposed group and 9% of the unexposed group had progressed. Antibiotic use was not associated with melanoma progression (HR: 0.81; 95% CI: 0.57–1.14). However, antibiotic use in subsequent 1 month and subsequent 3 months was associated with 31% reduction (HR: 0.69; 95% CI: 0.51–0.92) and 32% reduction (HR: 0.68; 95% CI: 0.51–0.91) in progression, respectively. In this cohort of patients with likely early-stage melanoma cancer, antibiotic use in 1 month and 3 months after melanoma surgery was associated with a lower risk of melanoma progression. Future studies are warranted to validate the findings.

## 1. Introduction

Melanoma ranks fifth and sixth among cancers in men and women, respectively, in the United States [1]. The incidence rate of melanoma increased from 22.2 per 100,000 persons in 2009 to 23.6 per 100,000 persons in 2016 in the US [2]. This increase in overall incidence rate is a result of an increase in new cases in older population, as melanoma incidence declined among adolescents and young adults between 2006 and 2015 [3]. The melanoma mortality rate slightly increased from 2.8 per 100,000 in 2009 to 3.1 per 100,000 in 2016 [2]. The 5-year melanoma-related survival rates for stage I and II melanoma were 97–99% and 82–94%, varying slightly based on the stage subgroup [4,5]. The melanoma-specific survival rate for stage III was 32–93%, markedly differing depending on the subgroup [4]. A total of 4% of incident malignant melanomas are metastatic (stage IV) at the time of diagnosis [6,7], which has 3-year overall survival rates of 5% to 26% depending on the organs involved [8].

Surgery is the first line of treatment for stages I through III melanoma, which is more effective when the tumor is localized and thin [9]. Wide local excision is the most common form of surgery, with 95% of melanoma surgeries using this technique [10]. The risk of recurrence is high when the tumor is thick, ulcerated, or spread to the lymph nodes [9]. The melanoma recurrence rates for stage IB, II, and III are 8%, 29%, and 47%, respectively, over a median follow-up of 5.4 years [11]. Immunotherapy drugs, such as pembrolizumab, nivolumab, and ipilimumab, are the recommended adjuvant therapies for stage III per the 2018 National Comprehensive Cancer Network (NCCN) guidelines [9]. Immunotherapies are highly effective in the metastatic setting as well, both as a single agent or in combination with another immunotherapy drug [12,13,14]. Chemotherapy has not been effective as adjuvant therapy for stage III patients and is not recommended [9]. Radiation therapy may still be used as adjuvant therapy and palliative treatment [9]. 

Antibiotic use is common in the US, although decreasing slightly in more recent years from 877 prescriptions per 1000 persons in 2011 to 836 per 1000 persons in 2016 [15]. Prescribing antibiotics outside of guidelines is a cause for concern, as it could lead to antibiotic resistance [16,17,18]. Broad-spectrum antibiotic use also interferes with the gut flora, which is responsible for mediating immune responses in the human body [19]. This could lead to breakdown of immune homeostasis in the gut, likely increasing the risk of incidence and progression of cancers, such as lung cancer, breast cancer, and melanoma [20,21,22]. Another study on leukemia and lymphoma reported that patients receiving anti-Gram-positive antibiotics had a 200% increase in the risk of death compared with those without antibiotic use [23]. Several recent studies have found that antibiotic usage 1 month prior to or within the 1st month of immune checkpoint inhibitor initiation negatively impacts the clinical outcomes in non-small cell lung cancer, renal cell cancer, urothelial cancer, and melanoma patients with advanced cancer [24,25,26,27]. However, it is unknown whether this negative association applies to melanoma patients in general or is specific to immune checkpoint inhibitors as they affect the immune system directly. On the other hand, several antibiotic agents, such as actinomycin and doxorubicin, have antitumor activity [28]. A study on melanoma cell-line culture showed that antibiotics that inhibit mitochondrial biogenesis, such as tetracyclines and chloramphenicol, have antineoplastic property [29]. 

In light of these mixed findings, our study aimed to study the association between antibiotic use and melanoma progression in a large administrative dataset of mainly commercially enrolled population. 

## 2. Methods

### 2.1. Data

We used a 10% random sample of IQVIA PharMetrics^®^ Plus data (January 2008–June 2018), a nationally representative administrative claims database of a primarily commercially insured population in the US. The database also contains claims from enrollees in Medicare Advantage, Medicaid managed care, and self-insured groups. All data are compliant with the Health Insurance Portability and Accountability Act (HIPAA) to protect patient privacy. The study was deemed a nonhuman subject research by the Institutional Review Board (IRB) at the University of Arkansas for Medical Sciences (UAMS) (IRB Number: 261726). 

### 2.2. Study Design and Patient Population

We conducted a retrospective cohort study. We identified patients with at least one diagnosis of malignant melanoma between 1 January 2009 and 30 June 2017 using the ICD-9-CM and ICD-10-CM diagnosis codes [30]. Since diagnosis codes may be used for diagnostic procedures for suspected melanoma in the claims database, we further required these patients to have undergone either wide local excision or Mohs micrographic surgeries within 90 days of diagnosis, which indicated confirmed melanoma diagnosis. This 90-day window was chosen because an exploratory analysis of our data showed that 97% of the patients who had undergone surgery within 1 year of melanoma diagnosis (57% of melanoma patients) did so within the first 90 days. This is also consistent with findings from another study [31]. Surgeries were identified using Current Procedural Terminology (CPT) codes, and the surgery date was defined as the index date. Patients were excluded if they had other treatments (chemotherapy, immunotherapy, targeted therapy, or radiation therapy), diagnosis of other cancers, or metastasis prior to or on the index date. Treatments were identified using level II HCPCS codes. Patients were also excluded if they had superficial or complete lymph node removal (lymphadenectomy) or sentinel lymph node excision on the index date, which resulted in a cohort of likely stage 1 patients. Patients were required to have 12 months of continuous insurance plan enrollment prior to the index date to identify history of comorbidities. This 12-month period prior to the index date was considered the baseline period. We excluded patients with HIV, organ transplant, rheumatoid arthritis, scleroderma and systemic lupus erythematosus, or other cancers in the baseline period. Three patient cohorts were constructed for the three time windows for checking antibiotic exposure. Patients were followed from the index date for a maximum of 2 years until progression, end of health insurance enrollment, or end of data availability (30 June 2018). For the analyses using the post-1-month and post-3-months windows, patients were required to be continuously enrolled for at least 1 month and 3 months after the index date, respectively. Therefore, their follow-up periods started from the end of the 1st month and 3 months, respectively. This approach was used to mitigate immortal time bias [32]. The study design is outlined in Figure 1. All diagnosis and procedure codes are included in Appendix A. 

### 2.3. Antibiotic Exposure

Antibiotic exposure was determined in three time windows: (i) 3 months prior to the index date, (ii) 1 month after and including the index date, and (iii) 3 months after and including the index date. We chose these time periods to mimic previous studies on the association of antibiotic exposure with immune therapy use in melanoma patients [26]. The pre-3-months window was selected also based on the clinical judgment that sub-clinical or clinical melanoma would be present during this time window. The post-1-month time window was selected based on its closeness to the melanoma surgery and the potential for its effect on the progression. The post-3-months window was used to explore whether the association observed in the 1-month post period was consistent when the time window for antibiotic exposure was extended to 3 months. For each exposure window, the exposed group included patients with prescriptions for broad-spectrum antibiotics, while the unexposed group included patients who did not receive those antibiotic prescriptions in the same period. We identified broad-spectrum antibiotic users using Generic Product Identifier (GPI) codes in the pharmacy files (Appendix A). Broad-spectrum antibiotics were loosely defined at the class level using a classification from a drug information portal [33].

### 2.4. Outcome

Melanoma progression in 2 years after surgery was the primary outcome. The 2-year period was selected based on a previous study that reported 2-year disease free rates in melanoma patients [34] and also for sample size consideration. Because of lack of a clinical variable for progression in the claims data, we developed an algorithm based on plausible clinical scenarios to identify progression. A patient was considered progressed if he or she had received hospice care at any time or any of the following after a gap of 6 months in diagnosis or treatment: (i) use of chemotherapy, immunotherapy, targeted therapy, or radiotherapy; (ii) new melanoma surgery; (iii) metastasis diagnosis; or (iv) diagnosis of a new primary cancer. The diagnosis of a new primary cancer was also considered for defining progression because a distant metastasis from primary melanoma could be coded as a nonmelanoma primary cancer, and including it has been shown to increase the sensitivity of metastasis identification [35]. Progression date was defined as the first date when any of these events occurred in 2 years following the index date. Patients’ time to progression was censored for those who were not considered to be progressed at the end of the second year, end of enrollment, or end of data availability, whichever was earliest. 

### 2.5. Covariates

We used prespecified covariates of age, sex, geographic region, and comorbidity (diabetes mellitus, liver disease, chronic kidney disease, inflammatory bowel disease, cardiovascular diseases, and chronic obstructive pulmonary disease). Previous literature has shown that these variables could increase the risk of melanoma [36]. Comorbid conditions were identified using diagnosis codes. In addition, we used a high-dimensional propensity score variable selection method to select 200 additional covariates for adjustment (see detailed description below) [37].

### 2.6. Statistical Analysis

A high-dimensional propensity score approach was used for covariate assessment. Following the approach of Schneeweiss et al., we used baseline information from pharmacy, inpatient diagnosis, outpatient diagnosis, inpatient procedure, and outpatient procedure dimensions [37]. Three-digit ICD-10-CM codes were used for identifying inpatient and outpatient diagnoses. For data prior to 1 October 2015, during which ICD-9-CM codes were used, ICD-9-CM codes were converted to ICD-10-CM codes using a published mapping algorithm [38]. Level I and level II HCPCS codes were used for inpatient and outpatient procedures. A six-digit GPI code, which identifies unique drug subclasses, was used from pharmacy data. We selected the top 200 binary covariates with the highest likelihood of bias based on the calculation outlined in Schneeweiss et al. [37]. These 200 covariates along with the prespecified covariates were used to generate the probability of antibiotic use (propensity score) using logistic regression. High-dimensional variable selection and generation of propensity score for antibiotic use was performed separately for the exposure measures using the three time windows. We used inverse probability treatment weighting to construct pseudo populations of antibiotic users and nonusers that were similar in the covariates [39]. We calculated the average treatment effect on the treated (ATT) using these weighted populations. We truncated the propensity score at 1st and 99th percentiles to prevent generation of large and unstable weights to improve precision [40]. We used weighted Cox proportional hazard regression for time to melanoma progression with antibiotic use as the exposure of interest and the only variable in the model. 

Three separate analyses were conducted for exposure measured in the three different windows. The proportional hazard assumption was tested using an interaction term between the exposure status and natural logarithm of follow-up time and was not rejected in all three analyses (pre-3-months: *p* = 0.295; post-1-month: *p* = 0.318; post-3-months: *p* = 0.374).

We used a *p*-value of ≤0.05 as the statistical significance threshold for all our analyses. SAS 9.4 was used for both descriptive and inferential analyses. 

### 2.7. Sensitivity Analyses

We conducted multiple sensitivity analyses.

Sensitivity Analysis 1: We assessed two alternative strategies for ascertaining melanoma diagnosis: (1) included patients who had surgery within 15 days of melanoma diagnosis, and (2) included patients who had surgery on the same day of the diagnosis. The same algorithm was applied to determine progression, and the same approach was used for the statistical analyses. 

Sensitivity Analysis 2: We examined whether lack of access to healthcare could explain the difference in progression between antibiotic users and nonusers since our progression definition was based on health encounter information. To do so, we regressed a variable for any healthcare encounters in 2 years post-surgery (inpatient, outpatient, emergency department, or prescription fills) on the exposure variable. If antibiotic exposure is associated with differential probability of having healthcare encounters following the surgery, it is an indication that the relationship between antibiotic use and progression may be an artifact of differential utilization of healthcare. 

Sensitivity Analysis 3: We conducted a falsification test. In a falsification test, one would use an outcome that is not likely to be affected by the exposure (“false outcome”) [41]. If a statistically significant association was discovered between the false outcome and the exposure, then we could not rule out the possibility that the association between antibiotic exposure and progression was due to chance or unobserved confounder instead of a real effect. On the other hand, if no association could be detected with this false outcome, then it provides confidence that the association we found was likely an unbiased one. For this test, we used a composite outcome of any chronic pain conditions (arthritis, back pain, neck pain, headache, or neuropathic pain). The ICD-9-CM and ICD-10 diagnosis codes were used to identify the pain conditions. While patients may experience chronic pains after surgery, it is unlikely that antibiotic use has a causal relationship with the pain conditions [42]. High-dimensional propensity score and inverse probability weighted Cox regressions were performed for any healthcare use and pain outcomes.

Sensitivity Analysis 4: We tested the stability of the results to alternative approaches of using the high-dimensional propensity scores: (1) we used stabilized weights instead of truncating the propensity scores, and (2) we adjusted for the propensity score in the Cox regression with the exposure group (regression adjustment) instead of inverse weighting. 

## 3. Results

Figure 2 presents the patient selection flow diagram. After applying the inclusion and exclusion criteria, a total of 3930 patients remained in the sample. This sample was used for the pre-3-months exposure analysis. For the post-1-month and post-3-months exposure analyses, we additionally required at least continuous 1-month and 3-month enrollments after surgery, respectively, to allow ascertainment of antibiotic exposure during these periods, which resulted in slightly smaller sample sizes (3831 for 1-month enrollment and 3587 patients for 3-month enrollment). 

Table 1 reports patients’ characteristics. The average age across the three samples was 56 years. Fifty-one percent were male, and 19–20% had the diagnosis in 2009. Four percent had chronic kidney disease, 12% had diabetes mellitus, and 9% had cardiovascular diseases in all the three samples. The percentages of patients with antibiotic use in the three exposure windows were: 16% (pre-3-months), 22% (post-1-month), and 22% (post-3-months). Comparing across the three time periods, a higher proportion of patients were exposed to antibiotics in the post-1-month and post-3-months periods compared with that in the pre-3-months period, although direct statistical testing was not feasible due to differences in sample sizes. Within each exposure window analysis, exposed and unexposed patients had similar distributions of age and year of diagnosis. In the pre-3-months window, 52% of the unexposed patients were male, while only 44% of the exposed were male (*p* < 0.001), and 5% of the unexposed had chronic obstructive pulmonary disease (COPD) compared with 8% of the exposed patients (*p* = 0.001). No statistically significant differences in sex or COPD prevalence were observed between the exposed and unexposed patients in the other time windows.

Table 2 and Table 3 report findings from the main analysis. In the pre-3-months analysis, 9% in the exposed group compared with 9% in the unexposed group progressed in the 2-year follow-up. After adjusting for difference in follow-up time, the incidence rates of progression in the exposed and unexposed groups were 0.19 and 0.19 per 1000 person-days, respectively (69.35 and 69.35 per 1000 person-years, respectively) (Table 2). The use of antibiotics was not associated with melanoma progression in both unadjusted (hazard ratio (HR): 1.04; 95% confidence interval (CI): 0.78–1.38) and adjusted Cox regression analyses using propensity weighting (HR: 0.81; 95% CI: 0.57–1.14). In the analysis for the post-1-month cohort, 8% in the exposed group and 9% in the unexposed group had progression. The incidence rates of progression in the exposed and unexposed groups were 0.13 and 0.12 per 1000 person-days, respectively (47.45 and 43.80 per 1000 person-years, respectively) (Table 2). In the adjusted Cox regression analysis using propensity weighting, use of antibiotics in the post-1-month cohort was associated with a 31% (HR: 0.69; 95% CI: 0.51–0.92) and 32% (HR: 0.68; 95% CI: 0.51–0.91) reduction in the risk of melanoma progression. 

Table 4 reports results from Sensitivity Analyses 1–3. In Sensitivity Analysis 1, using the two alternative strategies for ascertaining melanoma patients by restricting to patients with melanoma surgery on the same day as the diagnosis or within 15 days of diagnosis, antibiotic use in each of the three cohorts was not significantly associated with progression, although the point estimates for post-1-month and post-3-months were similar to the main analysis. In Sensitivity Analysis 2, antibiotic use in all the three cohorts was modestly associated with any healthcare use outcome, indicating marginally higher rates of healthcare utilization in the antibiotic users. In Sensitivity Analysis 3, which included a falsification test, none of the antibiotics exposure measures were associated with the composite outcome of chronic pain (HR and 95% CI: pre-3-months, 1.02 (0.90–1.16); post-1-month, 1.08 (0.97–1.21); and post-3-months, 1.07 (0.95–1.19)). 

In Sensitivity Analysis 4, stabilized weights (HR: 0.67; 95% CI: 0.41–1.10) and propensity score adjustment (HR: 0.76; 95% CI: 0.55–1.04) had similar results for the pre-90-days exposure window. Results similar to propensity weighting were observed for the post-1-month and post-3-months exposure windows (Table 3). 

## 4. Discussion

In this study of nationally representative commercially insured melanoma patients who received surgery within 90 days of diagnosis, broad-spectrum antibiotic use was not found to adversely affect the risk of progression. In fact, antibiotic exposure in the 1-month and 3-months post melanoma surgery was found to be associated with reduced risk of progression following surgery. Although the association was not statistically significant when the analyses were restricted to surgery on the same day as the diagnosis or within 15 days of surgery, the point estimates were similar to that observed in the within-90-days surgery cohort but had wider confidence intervals due to smaller sample sizes. No significant association was observed between antibiotic use in 3 months prior to the surgery and melanoma progression. Antibiotic users had slightly higher use of any healthcare services after surgery than nonusers, indicating that the lower rates of progression in the antibiotic users was unlikely a result of lower rates of healthcare encounters. Melanoma progression was observed in 8–9% of patients in our study; a study using cancer registry data reported melanoma recurrence of 5–6% in stage Ib and stage IIa patients, respectively [34]. Antibiotic use was not associated with the outcome of chronic pain, suggesting that unmeasured confounding may not be explaining the association of antibiotics and melanoma progression.

Certain antibiotics, such as cephalosporins, sulfonamides, tetracyclines, and fluoroquinolones, have been shown to possess antineoplastic properties [29]. Cephalosporins increased the activity of the glutathione S-convertase enzyme—a detoxifying enzyme that is involved in the protection against oxidative stress and DNA damage and is the first line of defense against carcinogens—in the liver and kidney in rat models [43,44,45]. Sulfonamides arrested the division of melanoma cells at the G0 and G1 phases in mice models, inhibiting tumor growth [46]. They also inhibited carbonic anhydrase enzymes IX and XII, which are responsible for maintaining the pH gradient between the cancer cells and extracellular fluid, leading to cancer proliferation and metastasis [47,48]. Tetracyclines inhibited mitochondrial protein biosynthesis, providing cytotoxic property to tetracycline compounds [49]. They also inhibited matrix metalloproteinases that are involved in tumor angiogenesis and metastasis [49,50]. Doxycycline reduced tumor burden in bone metastasis of breast cancer in mice models [51]. It disturbed cell homeostasis and induced DNA fragmentation, leading to apoptosis of melanoma cells in an in vitro study [52]. Fluoroquinolones have been shown to interfere with cell cycle, causing cell cycle arrest, and DNA fragmentation, precipitating cancer cell apoptosis [53]. Lomefloxacin arrested cell cycle likely due to topoisomerase II inhibition and also caused DNA breakdown at high doses in cultured melanoma human cells [54]. Chloramphenicol and its derivatives inhibited mitochondrial protein synthesis, killing the cancer stem cells [55,56]. Vancomycin has been shown to increase the activity of radiation therapy [57]. Although these multiple classes of antibiotics have been shown to possess antineoplastic activity, the evidence so far is limited to studies in animals or cancer cell lines. 

The human gut microbiome has a symbiotic relationship with the host immune system [58]. Animal studies have shown that the metabolites such as short-chain fatty acids and peptidoglycans produced by the gut microbes maintain local immunity in the intestinal region [58,59]. They aid in the proliferation and differentiation of helper T cells and also activate B cells for the production of immunoglobulin A (IgA) [58,59]. The role of the gut microbiome in the incidence of inflammatory bowel disease, asthma, depression, and dementia has been studied and well established: case-control-type studies have shown that the gut microbiota composition is different between healthy and diseased individuals [60,61,62,63]. Gut microbes are also involved in the pathogenesis of pancreatic cancer, hepatocellular carcinoma, and colorectal cancer [64,65,66]. A change in microbial composition has also been observed in patients with extraintestinal cancers, such as breast, prostate, and lung cancers [67,68,69]. A study on melanoma conducted using mice models reported that inoculating the mice with a *Bacteroides rodentium* strain led to the inhibition of melanoma growth [70]. When the mice were treated with antibiotics, the study reported that the inhibition of antitumor activity was halted, leading to increase in tumor progression [70]. Antibiotic treatment of mice with breast cancer also led to accelerated tumor growth [67]. 

Clinical studies assessing antibiotic use and cancer incidence have shown mixed findings [71]. A recent meta-analysis reported that antibiotic use was associated with increased risk of developing lung cancer (29%), lymphoma (31%), renal cell cancer (28%), and prostate cancer (25%), compared with no antibiotic use [71]. Likewise, a 14% increase in hazard of breast cancer was observed in antibiotic users in a study with 9 years of follow-up compared with no use [72]. Penicillins, fluoroquinolones, and tetracyclines are shown to have driven the risk of these cancers [71]. On the other hand, no statistically significant increase in the risk of melanoma was observed among antibiotic users compared with no users, while there was a 25% reduction in the incidence of cervical cancer [71]. Although interference of gut microbiota is considered the primary mechanism for this carcinogenic effect of antibiotics, methodological issues cast doubt on such findings. Bacterial infections could confound the association between antibiotics and cancer: infections by agents such as *Helicobacter pylori* are responsible for increasing cancer risk and also lead to the use of antibiotics [73]. The relationship between antibiotics and lung cancer was only observed among smokers, suggesting that confounding may be a big issue in the studies [73]. Additionally, individuals with subclinical cancer are at increased risk of bacterial infections and the subsequent use of antibiotics [73]. This suggests a reverse causal relationship between antibiotic use and cancer diagnosis.

Few studies have examined the association of antibiotic exposure with outcomes in cancer patients. In this study, we included likely early-stage-of-melanoma patients who received surgery as the first line of treatment for melanoma. For patients with advanced melanoma, immunotherapy has now become the mainstay treatment. Recent studies have found that antibiotic use prior to or around the time of immune checkpoint inhibitor initiation was associated with decreased survival and worse treatment outcomes in patients with advanced melanoma and other cancers [74,75,76,77,78]. However, no prior studies have evaluated such an association in early-stage-melanoma patients with surgery as the first line. Our study is not directly comparable with these studies. Our findings suggest that the negative association of immune checkpoint inhibitors and melanoma outcomes may not generalize to all melanoma patients. Future studies, especially controlled human studies, are needed to better understand the dynamics of antibiotics and cancer progression in melanoma.

This study is subject to several methodological limitations. First, it is a retrospective cohort study conducted using an insurance claims database. Although we used a high-dimensional propensity score method that adjusts for a large number of covariates, variables not captured in a claims database (family history of cancer, income status, tumor characteristics such as size, stage, and histology) could have confounded the results. The validity of the falsification test depends on whether the unmeasured confounding is similar for the study outcome and the falsification outcome, which may not have been the case in this study. Second, no direct information on progression is available from the database. We developed an algorithm for progression based on plausible clinical scenarios using information available in the database. However, the algorithm has not been validated and could lead to misclassification of cases and controls (patients who progressed and who did not, respectively). To the best of our knowledge, no validated claims-based progression algorithms have been developed for melanoma. However, a similar identification algorithm of disease progression and recurrence using administrative data has been studied and validated in breast cancer. For instance, similar to our approach, Xu et al. [79] used a second round of chemotherapy, a new breast cancer procedure, a new surgery, a new radiation therapy, or a second cluster of visits to oncologists, each after a gap of 180 days to identify breast cancer recurrence. The authors also tested gap periods of 365 days and 540 days; however, the increase in accuracy of the algorithm was minimal. Third, antibiotic use was identified using only outpatient prescriptions, and inpatient administration of antibiotics was not considered. Fourth, due to small sample sizes, an individual antibiotic class could not be assessed. We also did not assess the relationship between duration of antibiotic use and progression, as only a few patients with antibiotic use had progression. Future studies are warranted to evaluate the effect of type of antibiotics, cumulative use, and concomitant versus sequential use of different antibiotics on melanoma progression. Fifth, exposure to antibiotics was defined by the presence of prescriptions for broad-spectrum antibiotics in the three fixed time windows. It is possible that the nonexposure group could have been exposed to other narrow-spectrum antibiotics during these time periods or other time periods. Sixth, since death could not be ascertained in a claims database, termination of insurance enrollment could be due to death. This could lead to underestimation of progression, especially if death occurred earlier within 6 months of diagnosis and surgery. If disproportionally occurring more in the antibiotic-exposed group, this could potentially explain, at least partially, the lower risk of progression in that group. However, our progression algorithm included hospice use and/or a new round of cancer treatment as an indication of disease progression. Only death that was not preceded by hospice care or a new round of cancer treatment or due to other causes may be overlooked. Given the potentially early-stage melanoma patients included in our study and the extensive control of covariates using a high-dimensional propensity score method, this is unlikely, although still possible. 

## 5. Conclusions

Using a nationally representative commercial insurance claims database, our study showed that the use of broad-spectrum antibiotics was not associated with a higher risk of progression in patients with early-stage malignant melanoma who were treated with surgery as the first line of treatment. In contrast, antibiotic use within 1 month and 3 months following surgery was associated with lower risk of progression. Given the retrospective nature of the study with the data lacking clinical information on tumor characteristics and the low number of patients with progression in the exposed groups, further studies are needed to replicate and confirm the findings of this study.

## Figures and Tables

**Figure 1 cancers-13-04367-f001:**
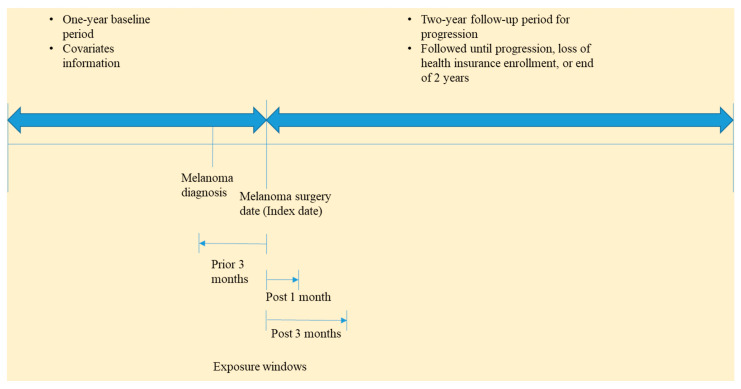
Study diagram for melanoma progression and antibiotic use.

**Figure 2 cancers-13-04367-f002:**
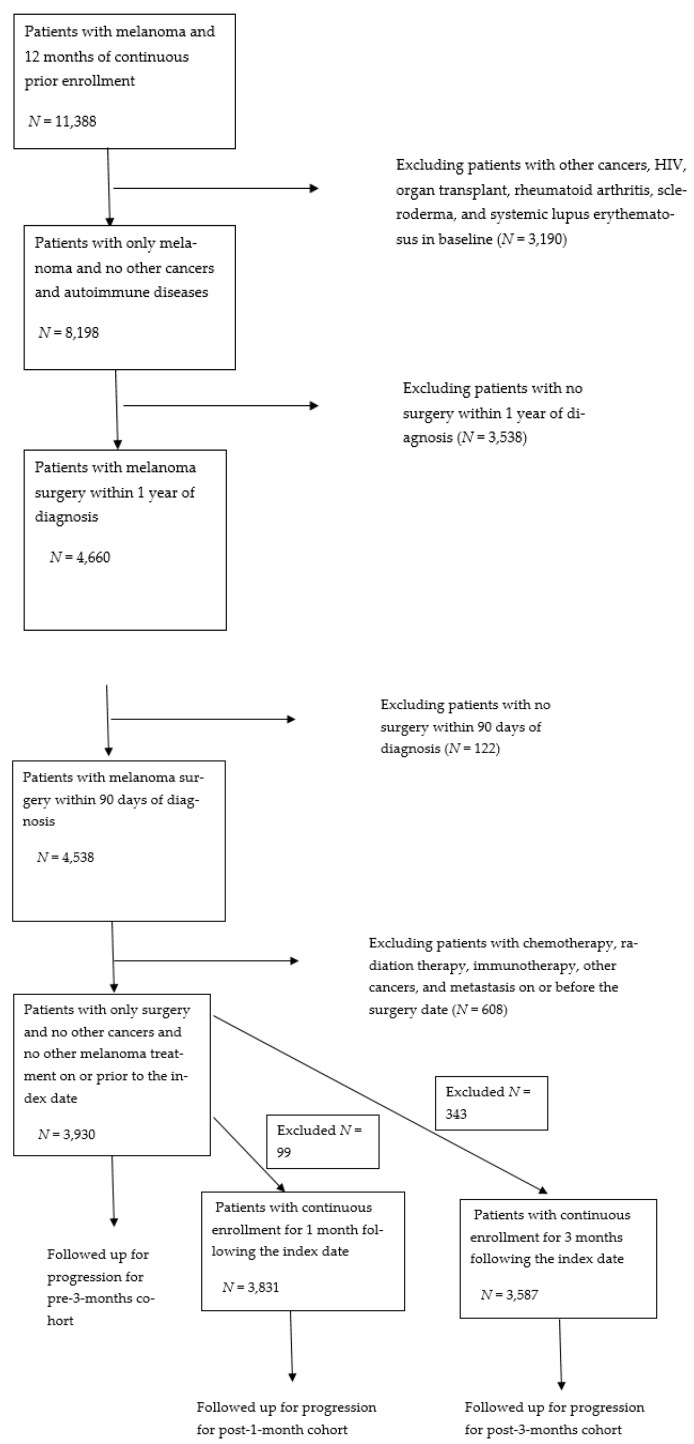
Flow diagram for sample selection.

**Table 1 cancers-13-04367-t001:** Demographic and comorbidity distribution by antibiotic exposure defined using the three time windows.

Characteristics	Antibiotics in Pre-3-Months; *n* (%)				Antibiotics in Post-1-Month; *n* (%)				Antibiotics in Post-3-Months; *n* (%)			
	Total (*n* = 3930)	No Use (*n* = 3284)	Use (*n* = 646)	*p*-Value	Total (*n* = 3831)	No Use (*n* = 3006)	Use (*n* = 825)	*p*-Value	Total (*n* = 3593)	No Use (*n* = 2819)	Use (*n* = 774)	*p*-Value
Age (years) mean (sd)	56.43 (13.78)	56.48 (13.72)	56.18 (14.09)	0.622	56.46 (13.77)	56.50 (13.67)	56.30 (14.15)	0.718	56.43 (13.83)	56.46 (13.69)	56.31 (14.34)	0.787
Gender				<0.001				0.883				0.831
Female	1926 (49.01)	1561 (47.53)	365 (56.50)		1872 (48.86)	1467 (48.80)	405 (49.09)		1753 (48.79)	1378 (48.88)	375 (48.45)	
Male	2004 (50.99)	1723 (52.47)	281 (43.50)		1959 (51.14)	1539 (51.20)	420 (50.91)		1840 (51.21)	1441 (51.12)	399 (51.55)	
Region				0.019				0.054				0.086
East	902 (22.95)	748 (22.78)	154 (23.84)		885 (23.10)	717 (23.85)	168 (20.36)		832 (23.16)	672 (23.84)	160 (20.67)	
Midwest	972 (24.73)	839 (25.55)	133 (20.59)		943 (24.61)	747 (24.85)	196 (23.76)		891 (24.80)	708 (25.12)	183 (23.64)	
South	1285 (32.70)	1047 (31.88)	238 (36.84)		1259 (32.86)	959 (31.90)	300 (36.36)		1179 (32.81)	899 (31.89)	280 (36.18)	
West	771 (19.62)	650 (19.79)	121 (18.73)		744 (19.42)	583 (19.39)	161 (19.52)		691 (19.23)	540 (19.16)	151 (19.51)	
Year of diagnosis				0.222				0.298				0.273
2009	743 (18.91)	628 (19.12)	115 (17.80)		732 (19.11)	597 (19.86)	135 (16.36)		706 (19.65)	577 (20.47)	129 (16.67)	
2010	658 (16.74)	543 (16.53)	115 (17.80)		646 (16.86)	505 (16.80)	141 (17.09)		624 (17.37)	488 (17.31)	136 (17.57)	
2011	591 (15.04)	493 (15.01)	98 (15.17)		582 (15.19)	448 (14.90)	134 (16.24)		557 (15.50)	427 (15.15)	130 (16.80)	
2012	532 (13.54)	429 (13.06)	103 (15.94)		509 (13.29)	389 (12.94)	120 (14.55)		455 (12.66)	352 (12.49)	103 (13.31)	
2013	327 (8.32)	276 (8.40)	51 (7.89)		321 (8.38)	250 (8.32)	71 (8.61)		302 (8.41)	235 (8.34)	67 (8.66)	
2014	290 (7.38)	243 (7.40)	47 (7.28)		280 (7.31)	220 (7.32)	60 (7.27)		250 (6.96)	195 (6.92)	55 (7.11)	
2015	306 (7.79)	250 (7.61)	56 (8.67)		297 (7.75)	223 (7.42)	74 (8.97)		280 (7.79)	208 (7.38)	72 (9.30)	
2016	333 (8.47)	289 (8.80)	44 (6.81)		323 (8.43)	261 (8.68)	62 (7.52)		296 (8.24)	239 (8.48)	57 (7.36)	
2017	150 (3.82)	133 (4.05)	17 (2.63)		141 (3.68)	113 (3.76)	28 (3.39)		123 (3.42)	98 (3.48)	25 (3.23)	
Chronic kidney disease	170 (4.33)	133 (4.05)	37 (5.73)	0.056	169 (4.41)	135 (4.49)	34 (4.12)	0.647	155 (4.31)	124 (4.40)	31 (4.01)	0.633
Diabetes mellitus	486 (12.37)	400 (12.18)	86 (13.31)	0.424	469 (12.24)	360 (11.98)	109 (13.21)	0.337	430 (11.97)	329 (11.67)	101 (13.05)	0.295
Cardiovascular Diseases	349 (8.88)	276 (8.40)	73 (11.30)	0.018	344 (8.98)	264 (8.78)	80 (9.70)	0.416	326 (9.07)	248 (8.80)	78 (10.08)	0.272
COPD	204 (5.19)	153 (4.66)	51 (7.89)	0.001	201 (5.25)	159 (5.29)	42 (5.09)	0.821	181 (5.04)	143 (5.07)	38 (4.91)	0.854
Liver disease	81 (2.06)	68 (2.07)	13 (2.01)	0.924	78 (2.04)	57 (1.90)	21 (2.55)	0.242	73 (2.03)	52 (1.84)	21 (2.71)	0.129
Inflammatory Bowel disease	33 (0.84)	25 (0.76)	8 (1.24)	0.224	32 (0.84)	22 (0.73)	10 (1.21)	0.179	30 (0.83)	21 (0.74)	9 (1.16)	0.258

**Table 2 cancers-13-04367-t002:** Progression rates: antibiotic use vs. no use by exposure window.

Outcomes	*n* (%)	Total Person-Days	n/1000 Person-Days	*n* (%)	Total Person-Days	n/1000 Person-Days
***Main analysis (surgery within 90 days)***						
	Antibiotics in Pre-3-Months					
	No Use (*n* = 3284)			Use (*n* = 646)		
Progression	282 (8.59)	1,524,248	0.19	56 (8.67)	292,890	0.19
	**Antibiotics in Post-1-Month**					
	**No use (*n* = 3006)**			**Use (*n* = 825)**		
Progression	272 (9.05)	2,106,838	0.13	68 (8.24)	578,860	0.12
	**Antibiotics in Post-3-Months**					
	**No use (*n* = 2819)**			**Use (*n* = 774)**		
Progression	271 (9.61)	1,970,534	0.14	68 (8.79)	541,630	0.13

**Table 3 cancers-13-04367-t003:** Cox proportional hazard regression results for melanoma progression.

Progression	Unadjusted HR (95% CI) for Antibiotic Use (Ref = No Use)	Adjusted HR (95% CI) for Antibiotic Use (Ref = No Use)
**Antibiotic use in Pre-3-Months**		
***Main Analysis***		
propensity score (PS) weighting	1.04 (0.78–1.38)	0.81 (0.57–1.14)
***Sensitivity Analysis 4***: Stabilized PS weights		0.67 (0.41–1.10)
PS adjustment		0.76 (0.55–1.04)
**Antibiotic use in Post-1-Month**		
***Main Analysis***		
propensity score (PS) weighting	0.91 (0.70–1.19)	0.69 (0.51–0.92)
***Sensitivity Analysis 4***: Stabilized PS weights		0.65 (0.48–0.88)
PS adjustment		0.69 (0.51–0.91)
**Antibiotic use in Post-3-Months**		
***Main Analysis***		
Propensity score (PS) weighting	0.91 (0.70–1.19)	0.68 (0.51–0.91)
***Sensitivity Analysis 4***: Stabilized PS weights		0.68 (0.51–0.91)
PS adjustment		0.65 (0.48–0.88)

PS: propensity score; HR: hazard ratio.

**Table 4 cancers-13-04367-t004:** Sensitivity analyses 1–3: antibiotic use vs. no use by exposure window.

Outcomes	*n* (%)	Total Person-Days	n/1000 Person-Days	*n* (%)	Total Person-Days	n/1000 Person-Days	Unadjusted Hazard Ratios (95% CI)	Adjusted (Propensity Weighting) Hazard Ratios (95% CI)
	Antibiotics in Pre-3-Months							
***Sensitivity Analysis 1: Alternative strategies for ascertaining melanoma cases***								
Same day surgery as diagnosis	**No use (*n* = 1087)**			**Use (*n* = 204)**				
Progression	92 (8.46)	519,124	0.18	17 (8.33)	97,032	0.18	0.99 (0.59–1.65)	0.68 (0.36–1.30)
Surgery within 15 days after diagnosis	**No use (*n* = 1986)**			**Use (*n* = 384)**				
Progression	168 (8.46)	938,653	0.18	32 (8.33)	181,702	0.18	0.99 (0.68–1.43)	0.72 (0.46–1.15)
***Sensitivity Analysis 2: Any healthcare encounter***	**No use (*n* = 3284)**			**Use (*n* = 646)**				
Any healthcare use	3217 (97.96)	114,797	28.02	643 (99.54)	10,785	59.62	1.37 (1.26–1.49)	1.14 (1.04–1.25)
***Sensitivity Analysis 3: Falsification test***	**No use (*n* = 3284)**			**Use (*n* = 646)**				
Chronic pain	1672 (50.91)	1,060,266	1.58	384 (59.44)	178,934	2.15	1.31 (1.17–1.47)	1.02 (0.90–1.16)
	**Antibiotics in Post-1-Month**							
***Sensitivity Analysis 1: Alternative strategies for ascertaining melanoma cases***								
Same day surgery as diagnosis	**No use (*n* = 1035)**			**Use (*n* = 229)**				
Progression	94 (9.08)	724,739	0.13	17 (7.42)	160,539	0.11	0.82 (0.49–1.38)	0.64 (0.37–1.12)
Surgery within 15 days after diagnosis	**No use (*n* = 1821)**			**Use (*n* = 498)**				
Progression	157 (8.62)	1,277,395	0.12	41 (8.23)	350,429	0.12	0.95 (0.68–1.34)	0.76 (0.52–1.09)
***Sensitivity Analysis 2: Any healthcare encounter***	**No use (*n* = 3006)**			**Use (*n* = 825)**				
Any healthcare use	2902 (96.54)	158,468	18.31	803 (97.33)	34,471	23.29	1.19 (1.10–1.29)	1.09 (1.01–1.19)
***Sensitivity Analysis 3: Falsification test***	**No use (*n* = 3006)**			**Use (*n* = 825)**				
Chronic pain	1538 (51.16)	1,465,928	1.05	451 (54.67)	368,602	1.22	1.17 (1.06–1.31)	1.08 (0.97–1.21)
	**Antibiotics in Post-3-Months**							
***Sensitivity Analysis 1: Alternative strategies for ascertaining melanoma cases***								
Same day surgery as diagnosis	**No use (*n* = 977)**			**Use (*n* = 213)**				
Progression	94 (9.62)	682,399	0.14	17 (7.98)	148,859	0.11	0.83 (0.49–1.40)	0.61 (0.35–1.06)
Surgery within 15 days after diagnosis	**No use (*n* = 1721)**			**Use (*n* = 462)**				
Progression	157 (9.12)	1,204,395	0.13	41 (8.87)	324,149	0.13	0.97 (0.69–1.37)	0.72 (0.49–1.05)
***Sensitivity Analysis 2: Any healthcare encounter***	**No use (*n* = 2819)**			**Use (*n* = 774)**				
Any healthcare use	2779 (98.58)	110,108	25.24	766 (98.97)	23,625	32.42	1.21 (1.11–1.31)	1.08 (1.00–1.18)
***Sensitivity Analysis 3: Falsification test***	**No use (*n* = 2819)**			**Use (*n* = 774)**				
Chronic pain	1508 (53.49)	1,349,985	1.12	444 (57.36)	336,359	1.32	1.19 (1.07–1.32)	1.07 (0.95–1.19)

Note: Results for Sensitivity Analysis 4—alternative propensity score methods are reported in Table 3.

## Data Availability

Data are not available upon request due to the proprietary nature of the data.

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
