# Peer review of "Broad-Spectrum Antibiotic Use and Disease Progression in Early-Stage Melanoma Patients: A Retrospective Cohort Study"

_cancers, 2021, doi:10.3390/cancers13174367_

Round 1

Reviewer 1 Report

The authors have extensively revised their manuscript and I don’t have any additional comments.

Reviewer 2 Report

The manuscript has been extensively revised. My remarks as well as the comments by the other two reviewer have been addressed appropriately and relevant new information has been incorporated in the manuscript. The revision has strengthed the presentation of findings. Limitations of the approach are clearly mentioned in the Discussion section.

Reviewer 3 Report

No further comments

This manuscript is a resubmission of an earlier submission. The following is a list of the peer review reports and author responses from that submission.

Round 1

Reviewer 1 Report

The authors studied the correlation between antibiotic use and disease progression in melanoma patients. With a cohort of 1291 patients, the data supported that antibiotic use was negatively correlated with cancer progression. The study is of interest to the broad cancer research field as it provided evidence that the use of antibiotic may delay the progression of certain cancer types. The manuscript is well written, and the data is interesting. There are several concerns:

1, As mentioned in Page 5, a patient was considered progressed under several situations, “including diagnosis of a new primary cancer”. The reason that diagnosis of a new primary cancer being considered a progress of melanoma should be provided.

2, As depicted in Page 7, patients with “no surgery on the same day as diagnosis” were excluded from this study, and a large amount of patient data were excluded (>6000). Is there any better way to make use of these excluded data to support the authors’ claim?

Author Response

Reviewer 1:

1, As mentioned in Page 5, a patient was considered progressed under several situations, “including diagnosis of a new primary cancer”. The reason that diagnosis of a new primary cancer being considered a progress of melanoma should be provided.

The diagnosis of new cancer was also considered for defining progression because the secondary malignancy diagnosis codes could miss certain metastasis that may be coded as a new primary cancer. We have added a citation that reported that using a primary cancer diagnosis to define metastasis in breast cancer patients increased the sensitivity. From page 9 in the manuscript:

The diagnosis of a new primary cancer was also considered for defining progression because a distant metastasis from primary melanoma could be coded as a nonmelanoma primary cancer, and including them has been shown to increase sensitivity of metastasis identification [35]

2, As depicted in Page 7, patients with “no surgery on the same day as diagnosis” were excluded from this study, and a large amount of patient data were excluded (>6000). Is there any better way to make use of these excluded data to support the authors’ claim?

Thank you for the excellent suggestion. In response to the reviewer’s comment, we have conducted exploratory analysis of the data. Of the 8,198 patients with a melanoma diagnosis, 4,660 patients had the surgery within one year of the first diagnosis. A total of 3,709 (80% of those who had surgery within one year) and 4,538 (97%) had the surgery within 30 days and 90 days of the diagnosis. A total of 29% had the surgery on the same day as the diagnosis. Based on this information, we have revised the study cohort selection to use 90-day as our main analysis and reported the surgery on the same day and within 15 days after diagnosis as the sensitivity analysis. Accordingly, the method section was revised as following (Page 4),

We identified patients with at least one diagnosis for malignant melanoma between Jan 01, 2009 and June 30, 2017 using ICD-9 CM and ICD-10 CM diagnosis codes [30]. Since diagnosis codes may be used for diagnostic procedures for suspected melanoma in the claims database, we further required these patients to have undergone either wide local excision or Mohr micrographic surgeries within 90 days of the diagnosis, which indicate confirmed melanoma diagnosis. This 90-day window was chosen because an exploratory analysis of our data showed that 97% of the patients who underwent surgery within one year of melanoma diagnosis did so within the first 90 days. This is also consistent with findings from another study [31].

Reviewer 2 Report

The manuscript addresses a potential effect of broad-spectrum antibiotic use on disease progression in (presumable) stage I melanoma patients based on health claims data. From a health claims database the authors construct a retrospective cohort study and elucidate the effect of antibiotic use on melanoma progression in a sophisticated analysis taking account of confounding using propensity score techniques. They find a strong protective effect of antibiotic use in a time interval of three months prior till one month after melanoma diagnosis/excision. Extensive sensitivity analyses corroborate the finding. A self-critical discussion points to limitations of the study.

The primary limitation of the study is the lack of access to clinical data of the study subjects. Using health claims data for a research question like the one studied by the authors is only a second choice option. However, the authors are fully aware of the problem and discuss the implications appropriately.

Specific remarks:

  • References in the Introduction to support epidemiological and clinical melanoma facts are quite old and should be replaced by references that are more recent.
  • l. 64/65: Stating that immunotherapeutic approaches are “costly with financial toxicity to the patients” shows that the authors have only the US audience in mind. In most European countries, patients do not feel the “financial toxicity” as their obligatory health insurances will cover these costs.
  • No reasons are given why only a 10% random sample of the health claims database is used as a basis for the analysis. Instead, at a later stage of the manuscript, the authors refer to the limited sample size as the reason for not performing more detailed analyses on the duration of antibiotic use and the type of antibiotics used. Why didn’t they use the full database to be able to have a large sample size for their analyses?
  • Please explain why inverse probability treatment weighting has been used to construct pseudo populations of antibiotic users and non-users, respectively, instead of incorporating the propensity score as an additional variable in the proportional hazard regression employed on the observed study population. What is the advantage of this “indirect” approach?
  • Has the assumption of the Cox model, i.e. proportional hazards of antibiotic users and non-users, been checked? The result should be reported in the manuscript.
  • Figure 1 should be graphically reworked or removed at all.
  • The algorithm to infer melanoma progression from health claims data has not been validated (l. 353-3257). Are there comparable studies for other cancers that used similar algorithms to defined progression based on health claims data? Please give references.

Overall, the manuscript makes the best from data that are not well suited to give a valid answer to the research question. The finding is interesting and warrants further investigation.

Author Response

Reviewer 2:

References in the Introduction to support epidemiological and clinical melanoma facts are quite old and should be replaced by references that are more recent.

We have updated the references in the first paragraph of the introduction to reflect more recent information. From the manuscript, page 3,

The incidence of melanoma increased from 22.2 per 100,000 population in 2009 to 23.6 per 100,000 in 2016 in the US [2]. This increase in overall incidence rates is the result of the increase in incidence in older population, as melanoma incidence declined for adolescents and young adults between 2006 and 2015 [3]. The melanoma mortality rates increased from 2.8 per 100,000 in 2009 to 3.1 per 100,000 in 2016 [2].

  1. 64/65: Stating that immunotherapeutic approaches are “costly with financial toxicity to the patients” shows that the authors have only the US audience in mind. In most European countries, patients do not feel the “financial toxicity” as their obligatory health insurances will cover these costs

We thank the reviewer for the comment. We have removed this sentence in the revised manuscript.

No reasons are given why only a 10% random sample of the health claims database is used as a basis for the analysis. Instead, at a later stage of the manuscript, the authors refer to the limited sample size as the reason for not performing more detailed analyses on the duration of antibiotic use and the type of antibiotics used. Why didn’t they use the full database to be able to have a large sample size for their analyses?

The dataset available for research at our institution is a 10% random sample of the IQVIA PharMetrics® Plus data. The full dataset is not available for research at our university, so we could only avail of the 10% of the total IQVIA PharMetrics® Plus data. The 10% sample of IQVIA PharMetrics® Plus data that we used has more than 10 million individuals who can be followed for a maximum of 10 years.

Please explain why inverse probability treatment weighting has been used to construct pseudo populations of antibiotic users and non-users, respectively, instead of incorporating the propensity score as an additional variable in the proportional hazard regression employed on the observed study population. What is the advantage of this “indirect” approach

The weighting approach provides the flexibility to estimate different treatment effects (average treatment effect, average treatment effect on the treated). We have cited a paper, which describes the advantages of weighting over adjustment (Desai et al., 2019; reference 39). Additionally, we have added two sensitivity analyses, where we 1) performed propensity score regression adjustment and 2) used stabilized weights. From page 11 of the manuscript,

We tested the stability of the results to alternative approaches of use of high-dimensional propensity scores: 1) We used stabilized weights instead of truncating the propensity scores, 2) We adjusted for the propensity score in the Cox regression with the exposure group (regression adjustment) instead of inverse weighting.

Has the assumption of the Cox model, i.e. proportional hazards of antibiotic users and non-users, been checked? The result should be reported in the manuscript.

We have added this information in the results section. From page 10 of the manuscript:

The proportional hazard assumption was tested using an interaction term between the exposure status and natural logarithm of follow-up time and was not rejected in all three analyses (prior 3-month: p=0.295; post 1-month: p=0.318; post 3-month: p=0.374).

Figure 1 should be graphically reworked or removed at all.

The figure 1 has been graphically reworked. We have kept it, as a graphical study design is recommended for retrospective studies (Schneeweiss et al, 2019).

The algorithm to infer melanoma progression from health claims data has not been validated (l. 353-3257). Are there comparable studies for other cancers that used similar algorithms to defined progression based on health claims data? Please give references.

We thank the reviewer for the comment. A similar approach has been used in breast cancer. We have provided the reference and also added a few sentences describing it. From the manuscript, page 16,

We developed an algorithm for progression based on plausible clinical scenarios using information available in the database. However, the algorithm has not been validated and could lead to misclassification of cases and controls (patients who progressed and who did not, respectively). To our best knowledge, no validated claims-based progression algorithms have been developed for melanoma However, similar identification algorithm of disease progression and recurrence using administrative data has been studied and validated in breast cancer. For instance, similar to our approach, Xu et al. [81] used a second round of chemotherapy, a new breast cancer procedure, a new surgery, a new radiation therapy, or a second cluster of visits to oncologists, each after the gap of 180 days to identify breast cancer recurrence.

Reviewer 3 Report

In this paper the authors study the association between antibiotics use and melanoma progression in in early stage patients only treated with surgery. The paper is clear and nicely written, but there is some methodological issues which is noted below. The study is also unfortunately hampered by the very low number of exposed patients experiencing the outcome of interest, progression.

Comments:

  1. Why is it important that the patient underwent surgery on the same day as diagnosis?
  2. Line 114: baseline period needs to be defined
  3. Why is the age not given for the post 3 months exposure in Table 1?
  4. Why are the number of patients decreasing over time (Table 1)? This should be commented on in the manuscript.
  5. What was the rationale for separating into those 3 exposure periods? This should be explained.
  6. How was use in those 3 periods defined, i.e. did you have exact date of the prescription? What did you do with prescriptions prescribed at the end of a period that would naturally continue into the next period?
  7. How was use in a different period accounted for in the Cox-regression, i.e. use in one month after diagnosis when analyzing 3 months after diagnosis?
  8. What does end of enrollment mean? How is it different from end of the two years of follow-up?
  9. Why were the patients only followed-up for 2 years? This should be explained in the manuscript.
  10. For propensity score, did the results differ if you stabilized the weights instead of truncating them?
  11. It is unclear how the pre-specified covariates where used. They did not seem to be adjusted for, but did they go into the propensity score?
  12. It does not seem that the authors preformed a time-dependent Cox-regression for antibiotics use for antibiotics use 1 and 3 months after diagnosis, this can induce immortal time bias, see for instance this paper by Suissa https://pubmed.ncbi.nlm.nih.gov/17252614/ and must be fixed.
  13. There is no mention of missing values. Was there really no missing values for any variables? This is very uncommon in observational studies. If there were missing values, how were these handled?
  14. I do not understand the basis of the falsification test. Chronic pain and melanoma progression is two very different outcomes, so in my opinion whether or not antibiotics use is associated with chronic pain (via a confounder) does not tell much about whether or not the association between antibiotics use and melanoma progression is confounded.
  15. You have a very low number of exposed patients with progression (4, 6 and 16 in the main analyses) this should be clearly stated in limitations and also mentioned in the conclusion.
  16. It should also be stated in the conclusion that these results are limited to early stage melanoma only treated with surgery.

Author Response

Reviewer 3:

Why is it important that the patient underwent surgery on the same day as diagnosis?

Thank you for the excellent subjection. In response to the reviewer’s comment, we conducted an exploratory analysis of the data. Of the 8,198 patients with a melanoma diagnosis, 4,660 patients had the surgery within one year of the first diagnosis. A total of 3,709 (80% of those who had surgery within one year) and 4,538 (97%) had the surgery within 30 days and 90 days of the diagnosis. A total of 29% had the surgery on the same day as the diagnosis. Based on this information, we have revised the study cohort selection to use 90-day as our main analysis and reported the surgery on the same day and within 15 days after diagnosis as the sensitivity analysis. Accordingly, the method section was revised as following (Page 4),

We identified patients with at least one diagnosis for malignant melanoma between Jan 01, 2009 and June 30, 2017 using ICD-9 CM and ICD-10 CM diagnosis codes [30]. Since diagnosis codes may be used for diagnostic procedures for suspected melanoma in the claims database, we further required these patients to have undergone either wide local excision or Mohr micrographic surgeries within 90 days of the diagnosis, which indicate confirmed melanoma diagnosis. This 90-day window was chosen because an exploratory analysis of our data showed that 97% of the patients who underwent surgery within one year of melanoma diagnosis did so within the first 90 days. This is also consistent with findings from another study [31].

Line 114: baseline period needs to be defined

We have added a sentence defining the baseline period. From the manuscript, page 4,

This 12-month period prior to the index date was considered the baseline period.

Why is the age not given for the post 3 months exposure in Table 1?

It was an oversight on our part. We have included it on the revised manuscript. We have revised the table 1 entirely by considering patients who underwent the surgery within 90 days of the diagnosis. Additionally for post 1 month and post 3 months cohorts, we required them to be continuously enrolled for 1 month and 3 months respectively following the surgery to reduce the risk of misclassification. Therefore, we have 3 separate samples for the three time windows in the revised manuscript.

Why are the number of patients decreasing over time (Table 1)? This should be commented on in the manuscript.

We believe the reviewer is referring to the decreasing number of unexposed patients (patients with no antibiotic prescription) defined using the three time windows. In the revised manuscript, we have constructed 3 separate samples for the three time windows because of additional inclusion criteria for post 1 and post 3 months cohorts. As a few patients are likely to not have continuous enrollment following the index date (surgery), the sample size decreased for the post 1 month cohort compared to the pre 3 months cohort and post 3 month cohort had the smallest sample size. We revised the description in Results on page 11:

Figure 2 is the patient selection flow diagram. After applying the inclusion and exclusion criteria, a total of 3,930 patients remained in the sample. This sample as used for the prior 3- month exposure analysis. For the post 1-month and post 3-month exposure analyses, we additionally required at least 1 month and 3 months continuous enrollment after surgery, respectively, to allow ascertainment of antibiotic exposure during these periods and thus, resulted in slightly smaller sample sizes (3,831 for 1-month and 3,587 patients for 3-month).

What was the rationale for separating into those 3 exposure periods? This should be explained.

The rationale for using the three time periods was to test whether use of antibiotics in the time window close to melanoma diagnosis has associations with the disease progression. The prior three months window was selected based on the clinical judgment that sub-clinical or clinical melanoma would be present during this time window. The post one month time window was selected based on its closeness to the melanoma surgery. The post three month window was used to explore if the association of post one month antibiotic use persisted if the time window for antibiotic use was extended to three months. We have added this information to the manuscript. From the manuscript, page 9,

We chose these time periods to mimic previous studies of the association of antibiotic exposure with immune therapy use in melanoma patients [26]. The prior three months window was selected also based on the clinical judgment that sub-clinical or clinical melanoma would be present during this time window. The post 1-month time window was selected based on its closeness to the melanoma surgery and the potential for its effect on the progression. The post 3-month window was used to explore if the association observed in the 1-month post period was consistent when the time window for antibiotic exposure was extended to three months.

How was use in those 3 periods defined, i.e. did you have exact date of the prescription? What did you do with prescriptions prescribed at the end of a period that would naturally continue into the next period?

Yes, we had exact dates of prescriptions. In the revised manuscript, we used the fill date of prescription and days’ supply to identify whether a patient was exposed in each of the exposure window or not. If a prescription prescribed in one time window would continue to another time window, then the patient was considered exposed in both time windows. For example, if a patient had an antibiotic prescription with 20 days’ supply that was prescribed 10 days prior to the surgery, the patient would be considered exposed in both pre 3 months and post 1 month windows.

How was use in a different period accounted for in the Cox-regression, i.e. use in one month after diagnosis when analyzing 3 months after diagnosis?

We conducted three separate analyses for the three exposure windows analysis. Since post 1-month and post 3-month periods overlap, for the post 3-month analysis, those who had exposure in the post 1-month period was considered as exposed in that analysis. In response to your comment below, we also implemented additional requirements of continued enrollment. Please see below the revised related description in the Methods on page 4.

For the analyses using the post 1-month and post 3-month windows, patients were required to be continuously enrolled for at least one month and three months after the index date, respectively. Therefore, their follow-up periods started from the end of the first month and three months respectively. This approach was used to mitigate immortal time bias [32].

What does end of enrollment mean? How is it different from end of the two years of follow-up?

This database is a commercial insurance claims database. So subjects can’t be followed further if their insurance coverage ended. This could happen before the end of the two-year period post surgery. The end of enrollment means the patients were no longer enrolled in the health insurance and their medical and pharmacy information is not available after that date. For instance, the patients could have lost the insurance after one year (365 days) of follow-up. These patients would be censored on 365th day of the follow-up and will not have full two year follow-up. All our analyses adjusted for differential follow-up of patients.

Why were the patients only followed-up for 2 years? This should be explained in the manuscript.

Given the longitudinal nature of the data from 2008 to 2018, some patients could theoretically be followed up for 9 years, while others may be followed up for less than 1 year. The 2-year maximum follow-up period was selected because we could compare it with the rates of melanoma progression in the existing literature, as a paper has reported 2-year melanoma-free rates. We have cited the paper in the manuscript. From the manuscript, page 9:

The two-year period was selected based on a paper that reported two-year disease free rates in melanoma patients [34] and also for sample size concerns.

For propensity score, did the results differ if you stabilized the weights instead of truncating them?

We have added the results when we stabilized the weights instead of truncating them in the table 3. The results were consistent. Please see now the Table 3 for results.

It is unclear how the pre-specified covariates where used. They did not seem to be adjusted for, but did they go into the propensity score?

Yes, they were used in the propensity score model in addition to the algorithm-derived covariates.  From the manuscript, page 9:

We used pre-specified covariates of age, sex, geographic region, and comorbidity (diabetes mellitus, liver disease, chronic kidney disease, inflammatory bowel disease, cardiovascular diseases and chronic obstructive pulmonary disease). Previous literature has shown that these variables could increase the risk of melanoma [39]….. In addition, we used a high-dimensional propensity score variable selection method to select 200 additional covariates for adjustment

It does not seem that the authors preformed a time-dependent Cox-regression for antibiotics use for antibiotics use 1 and 3 months after diagnosis, this can induce immortal time bias, see for instance this paper by Suissa https://pubmed.ncbi.nlm.nih.gov/17252614/ and must be fixed.

Thank you for this thoughtful suggestion. In response to your comment below, we also implemented additional requirements of continued enrollment. We required patients in post 1 month and post 3 months cohorts to have continuous enrollment for 1 and 3 months, respectively, following the index date. By doing so, we ensured the time zero of follow-up and also the date on which the exposure status was defined to be the same date, which avoids immortal time bias (Hernan et al., 2016; reference 32). Please see below the revised related description in the Methods on page 4.

For the analyses using the post 1-month and post 3-month windows, patients were required to be continuously enrolled for at least one month and three months after the index date, respectively. Therefore, their follow-up periods started from the end of the first month and three months respectively. This approach was used to mitigate immortal time bias [32].

There is no mention of missing values. Was there really no missing values for any variables? This is very uncommon in observational studies. If there were missing values, how were these handled?

As the study used healthcare claims data that has diagnosis and procedure codes for identifying disease and surgery, lack of those codes entails absence of the diagnosis or surgery. Although there is potential for misclassification (patients having melanoma but not having the diagnosis coded in their claims), the extent of misclassification could not be determined in the study, as this requires validation of diagnosis and procedure codes against information from health records.

I do not understand the basis of the falsification test. Chronic pain and melanoma progression is two very different outcomes, so in my opinion whether or not antibiotics use is associated with chronic pain (via a confounder) does not tell much about whether or not the association between antibiotics use and melanoma progression is confounded.

The falsification test was conducted under the assumption that antibiotic use would not be associated with the outcome of chronic pain after adjusting for the covariates. If a meaningful association was observed, it could indicate that unobserved confounding (such as socioeconomic status, access to care, among others) might be a problem. We also assumed that these same set of unobserved confounders could also confound the association between antibiotic use and melanoma progression. This assumption may not hold, so we have cautioned against over-interpretation of the falsification results in the manuscript page 16:

The validity of falsification tests depends on whether the unmeasured confounding is similar for the study outcome and the falsification outcomes, which may not have been the case in this study.

We also revised the description in the text as following on Page 10:

Sensitivity Analysis 3: We conducted a falsification test. In a falsification test, one would use an outcome that is not likely to be affected by the exposure (“false outcome”) [41]. If a statistically significant association was discovered between the false outcome and the exposure, then we could not rule out the possibility that the association between anti-biotic exposure and progression was due to chance or unobserved confounder instead of a real effect. On the other hand, if no association could be detected with this false outcome, then it provides confidence that the association we found was likely an unbiased one. For this test, we used a composite outcome of any chronic pain conditions (arthritis, back pain, neck pain, headache, or neuropathic pain). ICD-9 CM and ICD-10 diagnosis codes were used to identify the pain conditions. While patients may experience chronic pains after surgery, it is unlikely that antibiotics use has a causal relationship with the pain conditions. High-dimensional propensity score and inverse probability weighted Cox regressions were performed for any healthcare use and pain outcomes.

You have a very low number of exposed patients with progression (4, 6 and 16 in the main analyses) this should be clearly stated in limitations and also mentioned in the conclusion.

With the new main analysis of using surgery 90-day after diagnosis instead on the same day as diagnosis, these numbers increased. Also, we detected an issue with how we coded the progression algorithm, which also increased the number of patients with progression. Please see the revised Table 1 for results. The progression rate is comparable to a previous study. This information is added in the manuscript, page 15.

Melanoma progression was observed in 8–9% of patients in our study; a study using cancer registry data reported melanoma recurrence of 5–6% in stage Ib and stage IIa patients, respectively [34].

However, these changes did not affect the overall inferences materially. We are still observing a significant reduction in the risk of progression using the post 1-month and post 3-month exposure windows. We have revised the conclusion section of the manuscript accordingly (page 17):

Given the retrospective nature of the study with the data lacking clinical information on tumor characteristics and the lower number of patients with progression in the exposed groups, further studies are required to replicate and confirm the findings of this study.

It should also be stated in the conclusion that these results are limited to early stage melanoma only treated with surgery.

We have added this information to the conclusion section of the revised manuscript. From the manuscript, page 12,

, our study showed that use of broad-spectrum antibiotics was not associated with a higher risk of progression in patients with early stage malignant melanoma who were treated with surgery as the first line treatment. In contrast, antibiotic use within 1 month and 3 months following surgery was associated with lower risk of progression. Given the retrospective nature of the study with the data lacking clinical information on tumor characteristics and the low number of patients with progression in the exposed groups, further studies are needed to replicate and confirm the findings of this study.
